# Diagnosis of Glioblastoma by Immuno-Positron Emission Tomography

**DOI:** 10.3390/cancers14010074

**Published:** 2021-12-24

**Authors:** Eduardo Ruiz-López, Juan Calatayud-Pérez, Irene Castells-Yus, María José Gimeno-Peribáñez, Noelia Mendoza-Calvo, Miguel Ángel Morcillo, Alberto J. Schuhmacher

**Affiliations:** 1Molecular Oncology Group, Instituto de Investigación Sanitaria Aragón (IIS Aragón), 50009 Zaragoza, Spain; eruiz@iisaragon.es (E.R.-L.); icastells@iisaragon.es (I.C.-Y.); nmendoza@iisaragon.es (N.M.-C.); 2Department of Neurosurgery, Lozano Blesa University Clinical Hospital, 50009 Zaragoza, Spain; jbcalatayud@salud.aragon.es; 3Department of Radiology, Lozano Blesa University Clinical Hospital, 50009 Zaragoza, Spain; mjgimeno@comz.org; 4Biomedical Application of Radioisotopes and Pharmacokinetics Unit, Centro de Investigaciones Energéticas, Medioambientales y Tecnológicas (CIEMAT), 28040 Madrid, Spain; 5Fundación Agencia Aragonesa Para la Investigación y el Desarrollo (ARAID), 50018 Zaragoza, Spain

**Keywords:** diagnostic imaging, immuno-PET, glioblastoma, neuroimaging, molecular imaging, antibody, nanobody, theragnostic probes

## Abstract

**Simple Summary:**

Neuroimaging has transformed the way brain tumors are diagnosed and treated. Although different non-invasive modalities provide very helpful information, in some situations, they present a limited value. By merging the specificity of antibodies with the resolution, sensitivity, and quantitative capabilities of positron emission tomography (PET), “Immuno-PET” allows us to conduct the non-invasive diagnosis and monitoring of patients over time using antibody-based probes as an in vivo, integrated, quantifiable, 3D, full-body “immunohistochemistry”, like a “virtual biopsy”. This review provides and focuses on immuno-PET applications and future perspectives of this promising imaging approach for glioblastoma.

**Abstract:**

Neuroimaging has transformed neuro-oncology and the way that glioblastoma is diagnosed and treated. Magnetic Resonance Imaging (MRI) is the most widely used non-invasive technique in the primary diagnosis of glioblastoma. Although MRI provides very powerful anatomical information, it has proven to be of limited value for diagnosing glioblastomas in some situations. The final diagnosis requires a brain biopsy that may not depict the high intratumoral heterogeneity present in this tumor type. The revolution in “cancer-omics” is transforming the molecular classification of gliomas. However, many of the clinically relevant alterations revealed by these studies have not yet been integrated into the clinical management of patients, in part due to the lack of non-invasive biomarker-based imaging tools. An innovative option for biomarker identification in vivo is termed “immunotargeted imaging”. By merging the high target specificity of antibodies with the high spatial resolution, sensitivity, and quantitative capabilities of positron emission tomography (PET), “Immuno-PET” allows us to conduct the non-invasive diagnosis and monitoring of patients over time using antibody-based probes as an in vivo, integrated, quantifiable, 3D, full-body “immunohistochemistry” in patients. This review provides the state of the art of immuno-PET applications and future perspectives on this imaging approach for glioblastoma.

## 1. Introduction

Glioblastoma is the most common and aggressive tumor of the central nervous system in adults [1]. With an incidence of 3.23 cases per 100,000 individuals in Europe and the USA, glioblastoma represents ~49.1% of primary malignant brain tumors [1,2]. Despite continuous advances in the molecular classification of glioblastoma, and the steady progress in surgical, radiological, and chemotherapeutic treatment options [1,3,4], patient survival has improved only marginally during the past 3 decades. Current glioblastoma survival rates average just 8–14.6 months, with only ~5% of patients surviving more than 5 years [1,5]. Recurrence of glioblastoma is nearly universal and is associated with poor prognosis; patients with recurrent glioblastoma have a median survival of only 5–7 months with optimal therapy [6].

The current standard-of-care for treatment of newly diagnosed glioblastoma has remained relatively unchanged since 2005 and consists of maximal safe resection followed by concomitant chemoradiation with the alkylating agent temozolomide (TMZ), and subsequent adjuvant TMZ [7].

The DNA-repair enzyme O^6^-methylguanine-DNA methyltransferase (MGMT) impairs the killing of tumor cells by alkylating agents chemotherapy [8]. Methylation of the *MGMT* promoter regulates its expression. Despite confirming the prognostic significance of *MGMT* promoter methylation, survival did not improve with TMZ [9].

In 2011, a novel therapeutic approach, the first-generation tumor treating fields (TTF) device, was approved by the Food and Drug Administration (FDA) for the treatment of recurrent glioblastoma [10]. The TTF device was subsequently approved as adjuvant therapy for newly-diagnosed glioblastoma in 2015 [10,11].

Resistance to current treatments involves a complex interplay of numerous molecular mechanisms. Advances in the molecular classification of glioblastomas will likely translate into the development of novel and more effective therapeutic approaches that will improve glioblastoma patient outcomes.

## 2. Current Status of Glioblastoma Classification and Diagnosis

The 2016 World Health Organization (WHO) Classification of Tumors of the Central Nervous System (WHO CNS4) incorporated for the first time genetic alterations into the classification system to create more homogenous disease categories with greater prognostic value [12]. The WHO CNS4 classification symbolized a paradigm shift, replacing classical histology-based glioma diagnostics with an integrated histological and molecular classification system that enables more precise tumor categorization [12,13]. The incorporated diagnostic biomarkers in the 2016 WHO classification of gliomas were Isocitrate dehydrogenase (*IDH*)-*1/2* mutations, 1p/19q codeletion, H3 Histone, Family 3A (*H3F3A*) or HIST1H3B/C K27M (*H3-K27M*) mutations, and *C11orf95–RELA* fusions [13].

The novel 2021 classification (WHO CNS5) moves further to advance the role of molecular diagnostics in CNS tumor classification but stills remains rooted in other established approaches to tumor characterization, including histology and immunohistochemistry [14]. The WHO CNS5 assumes that most tumor types are aligned to distinct methylation profiles [15]. While these are not specified in every tumor definition, the information about diagnostic methylation is included in the “Definitions” and “Essential and Desirable Diagnostic Criteria” sections of WHO CNS5 and could provide more critical guidance for diagnosis [14].

WHO CNS5 considers all IDH mutant diffuse astrocytic tumors as “Astrocytoma, IDH-mutant” and are then graded as CNS WHO grade 2, 3, or 4. Furthermore, grading is no longer entirely histological, since the presence of CDKN2A/B homozygous deletion results in a CNS WHO grade of 4, even in the absence of microvascular proliferation or necrosis [14].

For a diagnosis of “Glioblastoma, IDH-wildtype” the novel WHO CNS5 incorporates 3 genetic parameters (*TERT* promoter mutation, *EGFR* gene amplification, combined gain of entire chromosome 7 and loss of entire chromosome 10) as criteria. For IDH-wildtype diffuse astrocytic tumors in adults, several works have shown that the presence of 1 or more of the 3 genetic parameters is sufficient to assign the highest WHO grade [16,17]. Consequently, “Glioblastoma, IDH-wildtype” in adults should be diagnosed in the setting of an IDH-wildtype diffuse and astrocytic glioma if there is either microvascular proliferation, or necrosis, or *TERT* promoter mutation, or *EGFR* gene amplification, or +7/−10 chromosome copy number changes. In IDH-wildtype diffuse astrocytomas occurring in younger age groups, however, consideration should be given to the different types of diffuse pediatric-type gliomas [14].

## 3. Neuroimaging

Neuroimaging has transformed neuro-oncology and the way glioblastoma is diagnosed and treated. First, with the advent of Computed Tomography (CT), and subsequently the Magnetic Resonance Imaging (MRI), these technologies have permitted an earlier identification of asymptomatic lesions. Nowadays, imaging is critical for pre-surgical diagnosis, intraoperative management, surgery, and ultimately monitoring after treatment with radiation and chemotherapy [18]. Anatomic imaging remains critical to identifying glioblastomas, but increasingly, advanced imaging methods allowing physiologic imaging have impacted the way these patients are managed [18].

### 3.1. Computed Tomography

Contrast agent-enhanced CT represented a major advance in modern neuroimaging, permitting an accurate anatomic localization of brain tumors and, by virtue of contrast enhancement, the malignant ones [19]. CT has wider availability, faster scanning times, and lower cost compared with MRI [20]. Despite these benefits, CT requires radiation exposure to the patient, which may be additive if serial imaging is needed. Furthermore, soft tissue resolution on CT is inferior to MR imaging [21]. The development of MRI diffusion-weighted sequences that allowed an indirect estimation of tumor cellularity transformed neuroimaging and replaced CT for the diagnosis of glioblastoma [19].

Still, CT can provide additional information regarding calcification or hemorrhage and be useful for subjects who cannot undergo MR imaging, such as in those with medical implants [18].

### 3.2. Magnetic Resonance Imaging

MRI is the most widely used, non-invasive technique in the primary diagnosis of brain tumors [2,3]. Although MRI provides very powerful anatomical information, it has proven to be of limited value for diagnosing gliomas in some situations. Specifically, MRI mainly provides detailed morphological information but does not effectively discriminate tumor tissue from concurrent processes such as inflammation, scarring, edema, or bleeding that can lead to incorrect estimates of the actual extension of the tumor mass (Figure 1) [3].

Furthermore, some lesions can be confounded with glioblastoma, including brain access (Figure 1), lumps, or space-occupying lesions (Figure 2a,b); certain demyelinating pathologies; the hemorrhagic transformation of stroke (Figure 2); or other lower-grade gliomas or brain tumors including metastases (Figure 3b).

The ability of conventional MRI to differentiate tumor tissue from post-therapeutic effects following neurosurgical resection, radiation (Figure 3a), alkylating chemotherapy, radiosurgery, and/or immunotherapy is also limited. For instance, the discrimination between tumor recurrence and post-surgical scar tissue is difficult to evaluate by MRI [3].

Although conventional MRI techniques are precious for evaluating the structure and anatomy of the brain, this may not be sufficient for the diagnosis of glioblastoma. Importantly, current improvements in MRI approaches can provide helpful physiological and functional information [22]. The emergence of novel MRI perfusion techniques offers enhanced procedures for tumor grading, guiding stereotactic biopsies, and monitoring treatment efficacy [23]. Perfusion imaging can help in treatment-related decision making, identify treatment-related processes (i.e., radiation necrosis, pseudoprogression, and pseudoregression), and be helpful to differentiate between tumor types and between tumor and non-neoplastic conditions [23].

Due to the limitations of MRI, the final diagnosis of glioblastoma requires a stereotactic brain biopsy and/or post-surgery histopathological analysis. However, glioblastomas present a high intra-tumoral heterogeneity, which undermines the use of a single biopsy for determining the tumor genotype, and has implications for potential targeted therapies [24]. Furthermore, biopsies might promote the natural history of glioblastomas [25,26]. These disadvantages highlight the importance of looking for other molecular-imaging-based technologies that allow efficient and safe diagnosis of these brain tumors.

### 3.3. Positron Emission Tomography (PET)

PET is routinely used in diagnosing, grading, and staging cancers and in assessing the efficacy of therapies. While MRI provides useful anatomical data, PET provides complementary biochemical information obtained in a non-invasive manner [27]. PET provides the highest sensitivity and resolution compared to other imaging modalities, allowing for the detection of very small tumors. The development of PET has had broad consequences in clinical practice and has been associated with an estimated change in intended management in about one-third of cases [28].

To date, many PET agents have been developed. The already-established PET tracers are focused on general cancer hallmarks [29] that are not specific to any tumor type. Most of them are sustained proliferation markers that indicate an increase in glucose metabolism, protein synthesis, or DNA replication. Thus, the PET tracers generally consist of radionuclide-labeled forms of the “building blocks” of macromolecules: sugars, amino acids, and nucleotide bases [27]. The gold standard tracer for most PET cancer imaging is 2-[^18^F]fluoro-2-deoxy-D-glucose ([^18^F]FDG), a fluorine-18 glucose analog [30], being the most widely used in clinical radiopharmaceutical practice, and accounting for more than 90% of total PET scans [31]. There are several tracers based on neutral amino acid analogues, such as [^11^C]methionine ([^11^C]MET), [^18^F]fluoroethyl-tyrosine ([^18^F]FET), [^18^F]L-fluoro-dihydroxyphenylalanine ([^18^F]FDOPA), or [^18^F]fluoro-thymidine ([^18^F]FLT), that show high diagnostic performance [27,32]. Another non-aminoacidic tracer that can be used in brain tumors is choline, either as [^11^C]choline or [^18^F]choline, as tumor cells increase choline uptake since they experience high phospholipid turnover in their membranes [32,33,34]. [^11^C]choline can be used for tumor monitoring, as it presents high diagnostic accuracy for the differentiation of glioma relapses from radiation-induced necrosis [35], and [^18^F]choline could be potentially used as a surgical imaging biomarker, as it was proven helpful for the discrimination of the highly proliferative peripheral area of the tumor and intraparenchymal hemorrhage [36].

The use of metabolic alteration tracers is helpful due to their simplicity, but metabolic changes are not unique to cancers. Hence, although useful in diagnosis, they are most appropriate for disease monitoring. For example, [^18^F]FDG is ineffective for diagnosing gliomas due to the high glucose metabolism in the normal brain, which results in suboptimal tumor detection and delineation, especially upon treatment [3,37]. Besides, many PET tracers present a limited blood-brain barrier (BBB) penetration that limits their use for glioblastoma imaging [27]. Other radiotracers for brain tumors are currently under evaluation, and include the glutamine analog 4-^18^F-(2S,4R)-fluoroglutamine, which shows high uptake in gliomas but low background brain uptake and may facilitate clear tumor delineation [38].

There are other tracers currently in use that focus on different processes than metabolism [27,39]. Multiple radiotracers sense oxygen levels, such as [^18^F]Fluoromisoinodazole ([^18^F]FMISO), which can be used to visualize hypoxia [40]. The success of [^18^F]FMISO in glioma imaging is limited for its low sensitivity differentiating normoxic and hypoxic tissue and its low BBB permeability [41,42]. Nevertheless, hypoxia imaging can be useful in solid tumors treatment since this biochemical process is critical for monitoring the effective regression after targeted radiotherapy [43].

Another process that is useful for imaging is inflammation, characteristic of glioblastoma pathology. Small molecule inhibitors of the mitochondrial membrane’s translocator protein (TSPO) are also used as PET tracers [44]. Their use in glioblastoma is limited due to the heterogeneity of the tumor and because the signal from tumor-related inflammation cannot be well distinguished from the signal caused by radiation therapy [27].

New PET imaging molecules are being developed hand in hand with new therapeutical advances that target proliferation, immunity, and genetic modifications. These new imaging agents use tumor-specific biomarkers rather than general ligands of proliferation, hypoxia, or inflammation. Some examples, yet to be used in human brain imaging, target sigma 1 (associated with invasiveness), sigma 2 (associated with proliferation), PD-L1 (immune checkpoint), epidermal growth factor receptor (EGFR), ADP-ribose polymerase (PARP), or isocitrate dehydrogenase (IDH), among others [27,45].

There remains an unmet need for highly specific imaging tools that allow for identifying brain tumors at early stages, monitoring changes upon treatment, and determining signs of progression or recurrence. An innovative and attractive alternative is termed “immunotargeted imaging” [46,47]. This approach merges the target selectivity and specificity of antibodies and engineered fragments toward a given tumor cell surface marker with PET imaging techniques to generate “immuno-PET”. By merging the high target specificity of antibodies with the high spatial resolution, sensitivity, and quantitative capabilities of PET, it is possible to conduct the non-invasive diagnosis and monitoring of patients over time using in vivo, integrated, quantifiable, 3D, full-body immunohistochemistry (IHC) as a “virtual biopsy”.

## 4. Elements of Immuno-PET: Target, Antibody and Radionuclide

We live within a “cancer-omics” revolution that reveals many clinically relevant alterations that are not yet included into the medical practice, at least partly due to the limited number of non-invasive imaging biomarkers [48]. An innovative option, termed “immunotargeted imaging”, merges the target specificity and selectivity of antibodies and derivatives towards a given tumor cell surface marker with the capabilities of a given imaging technique. Immunotargeted imaging by PET necessitates three components that are required to fulfill several characteristics: a suitable target for imaging, an optimally engineered antibody for imaging applications, and selecting an appropriate radionuclide for immuno-PET (Figure 4).

### 4.1. Selection of an Appropriate Target for Immuno-PET

A suitable epitope for immuno-PET needs: (I) to be exposed on the outer surface of the plasma membrane or to have extracellular components to facilitate the access of the antibody or a derivative, (II) to be highly expressed in the tumor, but (III) to be absent or to present low expression levels in healthy tissue. The target is not limited to malignant cells, it can also be related to different tumor components, including vasculature, stromal cells, and extracellular matrix and infiltrating immune cells [49].

Ideally, a biomarker should provide additional information about the tumor and predict prognosis, survival, or therapeutic outcome. In recent years, several pan-cancer studies have been performed [50]. The growing number of massive glioblastoma-specific databases containing multi-omics data (transcriptome, genome, epigenome, proteome, degradome, kinome, microbiome, metagenome, and metabolome, among others) linked to clinical data [51], together with the development of bioinformatics, facilitates the identification of novel biomarkers that could serve as bases to develop immuno-PET probes to target gliomas.

### 4.2. Selection of Optimally Engineered Antibody Derivates for Immuno-PET

Antibodies can recognize epitopes with high affinity and specificity. Conventional antibodies present an extended serum half-life that ranges from days up to 3 weeks. This long exposure is a suitable property for antibodies to be used as therapeutics. However, while the exposure of the antibody to the target would be incremented [46,49,52], this could represent a limitation for imaging, as they require several days for being cleared from blood and background for proper visualization by immuno-PET [53].

Ideal immuno-PET imaging requires a highly specific tumor uptake and low background retention of the radiotracer. To this end, a tracer should specifically bind and saturate its target as quickly as possible, and the unbound tracer should be cleared out rapidly from the blood. Protein engineering of antibodies allows the production of smaller fragments maintaining their antigen specificity and affinity with different pharmacokinetics. The clearance of these antibody derivatives can be influenced by their size, surface charge, and hydrophilicity/hydrophobicity, as well as any fused or conjugated molecules [46,49].

Removal of the Fc region results in a faster blood clearance and increased tumor-penetration rate, allowing imaging within several hours after injection [54]. The Fc fragment absence avoids its related specific functions including complement- and effector cell-mediated immune reactions. Moreover, the reduction of the size of antibodies favors its clearance through the renal system (whose protein clearance threshold is ~60 kDa).

In F(ab′)_2_ (F(ab′)_2_-Fab dimer) and Fab (Fragment antigen-binding) antibody fragments, the CH_2_ domain is removed, increasing their blood clearance, allowing a faster visualization and optimizing images. Although F(ab′)_2_ size (~120 kDa) is over the kidney size clearance threshold, both F(ab′)_2_ and Fab fragments are cleared through this pathway. This could be due to the enzymatic cleavage of F(ab′)_2_ into smaller molecules such as Fab, which can pass the glomerular membrane [55,56].

One of the most common formats is the single-chain variable fragment (scFv, ~25 kDa), which covalently binds a light chain variable domain (VL) with a heavy chain variable domain (VH) through a flexible peptide [60]. Importantly, this linker can be modified to allow cell permeability and BBB penetration [61].

scFvs can also be modified by adding different portions of the constant region (CH), generating different fragments. Diabodies consist of an scFv dimer (Db; ~50 kDa) connected by a short linker that does not permit to pair the two dimers in the same chain; they interact with the antigens in a divalent manner. Minibodies (Mb; ~80 kDa) are fusion proteins composed of an scFv fused to a single Fc [47,53]. The specificity for the antigen remains intact for all these fragments; they also have a better blood clearance and a better background-signal ratio. Moreover, in contrast to complete antibodies, these fragments could pass through the BBB more efficiently [52,62].

Nanobodies (Nb; ~15 kDa) are the single variable domain-heavy chain fragment (VHH) of the heavy-chain-only antibodies (HCAbs) derived from the *Camelidae* species [63]. Variable new antigen receptors (VNARs) derive from the single variable domain of the heavy-chain-only antibodies or immunoglobulin new antigen receptors (IgNARs) of cartilaginous fish [64]. Smaller size and increased plasticity allow nanobodies and VNARs to recognize unique conformational epitopes, including unstructured regions of intrinsically disordered proteins [65] and active sites of enzymes and cavities of receptors [66,67].

Affibodies (AB; ~7 kDa are nonimmunogenic three-helix scaffold-based derived peptides (~58 amino acids) engineered from the protein A of *Staphylococcus aureus,* which can recognize a wide variety of targets with high affinity) [68,69]. Owing to their small size they can access antigens that would be unattainable to conventional Fv-based antibodies and derivatives. However, it is crucial to consider that due to their small size the clearance will be faster than other bigger fragments [68,70]. To prevent unwanted immune responses induced by non-human antibodies and antibody fragment derivatives, these can be “humanized” by modifying their protein sequences [71,72,73].

### 4.3. Selection of an Adequate Radionuclide That Fits with the Engineered Antibody Derivates

The third component of the immuno-PET consists of the positron-emitting radionuclide. It is critical to match the biological half-life of the antibody or fragment being used with the physical half-life of the positron-emitting radionuclide. It is also essential that the obtained radioimmunoconjugates preserve their affinity to the target and preserve or improve their properties.

To successfully detect the antibody binding to the target by PET, it is necessary to have a positron-emitter linked to the antibody that has to be in a stable and inert way. For those immune molecules with a long circulating half-life (slow kinetics) such as intact conventional antibodies (t_1/2_ = days to weeks), radionuclides with longer half-lives such as ^89^Zr (t_1/2_ = 78.4 h) and ^124^I (t_1/2_ = 100.3 h) will be more suitable [74]. Smaller antibody-derived fragments can be labeled with intermediate half-life radionuclides such as ^64^Cu (t_1/2_ = 12.7 h) and ^86^Y (t_1/2_ = 14.7 h), or short half-life such as ^18^F (t_1/2_ = 110 min), ^44^Sc (t_1/2_ = 3.94 h), and ^68^Ga (t_1/2_ = 67.7 min). Due to its small size, affibodies could be combined with isotopes of shorter half-lives, such as ^18^F and ^99^mTc) [68].

Although ^68^Ga or ^44^Sc can be produced in a cyclotron [75,76,77,78], they can advantageously be produced in a cyclotron-independent manner [79,80,81]. These radionuclides can be produced in a commercially available ^68^Ge/^68^Ga or ^44^Ti/^44^Sc generator, a result more affordable and accessible to any PET center [79,80,81].

^44^Sc presents further advantages allowing multiplexed PET (mPET) imaging. ^44^Sc emits prompt gamma-rays right after the positron emission that can be distinguished from standard positron emitters such as ^18^F or ^68^Ga [80,81,82], enabling the simultaneous non-invasive imaging of two different radiotracers with PET scanners [81].

Radionuclides can be directly conjugated to the antibody or engineered form (such as ^18^F, ^124^I and ^76^Br by radiohalogenation [83]) or indirectly using a chelating moiety that serves as linkers. The linker contains a chelating group for the attachment of radiometals and a group that reacts with ε-amino groups of lysine residues and/or N-terminus of the antibody form. The most widely used for immuno-PET are siderophore desferrioxamine-B (DFO), hexadentate tris(hydroxamate), 1,4,7,10-tetraazacyclododecane-1,4,7,10-tetraacetic acid (DOTA), and 1,4,7-triazacyclononane-1,4,7-triacetic acid (NOTA), among others [84,85,86].

Direct and linker-mediated binding of the positron-emitter radionuclide to the antibody can incorporate the tracer at random sites in the protein altering of their antigen-binding site [46]. Further strategies have been designed to conjugate the linker in a site-directed manner to avoid this issue [46,84,85,86]. These alternatives include click chemistry by biorthogonal reactions between two coupling partners (i.e., alkyne and azide) [87].

Immuno-PET imaging requires simple, fast, and specific radiolabeling of antibody-based probes under mild conditions. Biorthogonal reactions fulfill these criteria, as they can display high selectivity and produce a chemically and biologically inert product/ linkage. These selective reactions are kinetically fast and biocompatible, occurring at physiological pH, temperature, and in a physiologically relevant solvent milieu. These quick and modular reactions give high product yields and remain physically stable [88]. Notably, several selective bioorthogonal reactions can occur in living systems, allowing for a two-step pretargeting strategy [89]. In this setting, a primed antibody or subunit, previously linked to one of the reaction components, can be administered prior to completion of the reaction [87,90,91]. Then, the second component (i.e., a linker/chelating agent marked with the radionuclide) can be administered hours to days later depending on the antibody format and clearance route. This strategy permits the use of smaller doses of radioactive material, provides a faster clearance and reduces patients’ exposure to radioactivity while providing a better signal-to-noise ratio.

Furthermore, the two-step pretargeting strategy allows the labeling of different tracers (e.g., fluorescent dyes for fluorescence-guided surgery, MRI-tracers such as (Gd)-complexes, or SPIO nanoparticles, among others) to the same pretargeted antibody to generate multi-modal and/or multifunctional imaging agents [92,93].

## 5. Current Perspectives of Immuno-PET for Glioblastoma

Several targets are functionally relevant in glioblastoma, since they have clinical potential as prognostic markers. In addition, they could be used as molecular targets for the delivery of agents for their detection. To date, immuno-PET imaging probes have been mainly designed to target glioblastoma tumors in preclinical models. Several of them have already been successful in detecting gliomas in preclinical studies, as shown in Table 1. These tracers allow for evaluating multiple hallmarks [29] of gliomas and the treatment response in preclinical settings.

Several immuno-PET tracers’ [94,95,96,97,98,99,100,101,106,107] target membrane proteins whose expression is altered in glioblastoma including the Epidermal Growth Factor Receptor (EGFR), Delta-Like Ligand 4 (DLL4), Ephrin type-A receptor 2 (EPHA2), Cluster of differentiation 47 (CD47), the AC133 antigen, and the Membrane-type 1 matrix metalloproteinase (MT1-MMP/MMP14) (Figure 5). In vivo administration of these tracers showed high-specific-contrast imaging of the target in an MT1-MMP expressing glioblastoma tumor model and provided strong evidence for their utility as an alternative to non-specific imaging of glioblastoma.

Glioblastomas develop in complex tissue environments, which support sustained growth, invasion, progression, and response to therapies [117]. Several components of the tumor microenvironment such as vessels [108,109,110], macrophages, and extracellular matrix proteins [104,105] are also promising candidates for the development of immuno-PET diagnostic approaches in glioblastoma [108,109,110,114].

Re-education of the tumor microenvironment of glioblastomas emerges as a novel opportunity for therapeutic intervention, as it has anti-tumorigenic effects [118,119].

Macrophages and microglia accumulate with glioblastoma progression and can be targeted via inhibition of Colony-Stimulating Factor-1 Receptor (CSF-1R) to regress high-grade tumors in animal models of glioblastoma [118,119]. A recent immuno-PET tracer targeting the Integrin αM (CD11b) expressing cells (macrophages) with high specificity in a mouse model of glioblastoma was developed, demonstrating the potential for non-invasive quantification of tumor-infiltrating CD11b+ immune cells during disease progression and immunotherapy in patients suffering of glioblastoma [99,114]. Another anti-CD11b tracer has been shown to be effective in mouse models for imaging tumor-associated myeloid cells (TAMCs), which constitute up to 40% of the cell mass of gliomas [115].

Immunotherapy, especially immune-checkpoint inhibitors, is transforming oncology. Despite glioblastomas frequently express the programmed cell death ligand 1 (PD-L1), the results obtained with anti-PD1 therapy are below expectations. The frequent intratumor variability of PD-L1 expression carries significant implications for determination accuracy. PET imaging of immune-checkpoint inhibitors may serve as a robust biomarker to predict and monitor responses to these immunotherapies, complementing the existing immunohistochemical techniques [120].

Other immuno-PET tracers targeting immune cells have been evaluated. A tracer targeting CD8+ T cell immune response to oncolytic herpes simplex virus (oHSV) M002 immunotherapy was evaluated as a proof of concept in a syngeneic glioblastoma model [113]. Another monoclonal antibody-based tracer was developed for immuno-PET imaging of T-cell activation targeting the costimulatory receptor OX40, and used to monitor the stimulated T-cell response in a murine orthotopic glioma model [116].

Furthermore, some of these immuno-PET tracers are valuable tools to determine the transient BBB disruption and permeability induced by mannitol [102] or produced by the combination of injected microbubbles with low-intensity focused ultrasound in vivo [97,103,111]. Notably, [^89^Zr]Zr-DFO-fresolimumab, an immuno-PET tracer based on a monoclonal antibody that can neutralize all mammalian isoforms of TGF-β, was assayed in humans and penetrated recurrent high-grade gliomas (Figure 5c) but did not result in clinical benefit [109].

## 6. Novel Nanobody-Based Immuno-PET Imaging Methods for Glioblastoma

The development of immuno-PET probes for the diagnosis of glioblastoma may encounter several hurdles to be reached due to the intracranial location of this tumor type. CNS barriers may limit the delivery of conventional antibody-based immuno-PET probes. The restricted entrance of molecules into the CNS is exerted mainly by the blood–brain barrier (BBB) and the blood–cerebrospinal fluid (CSF) barrier (BCSFB) [121]. These dynamic interfaces allow the exclusive passage from the blood into the CNS of receptor-specific ligands and small molecules (MW < 400 Da) that are lipid-soluble [122,123]. The delivery of peptide and protein drugs through the BBB is a major challenge for treating CNS diseases, and strategies to achieve therapeutic concentrations are under development [124]. In this regard, only 0.01–0.4% of the total amount of administered therapeutic antibodies have access to the CNS through passive diffusion [125,126]. Transport of therapeutic antibodies, mostly with the IgG isotype (150 kDa), may be hampered by the binding of their Fc domain to Fc receptors in the BBB [127]. Both the Fcγ receptor (FcγR) and neonatal Fc receptor (FcRn) have been implicated in the inverse transport of IgG through the BBB and their subsequent return from the brain to blood circulation [128,129]. Nevertheless, recent studies have proposed that antibody transcytosis across the BBB is carried by non-saturable, non-specific, Fc-independent mechanisms [130]. These mechanisms may hinder the diagnostic potential of monoclonal antibody-based immune-PET tracers for glioblastoma patients.

The development of antibody subunits targeting glioblastoma biomarkers that overcome the BBB selectivity emerges as a promising tool that could contribute to glioblastoma diagnosis by immuno-PET [131]. Single-domain antibodies (sdAbs) such as nanobodies have a lower MW, enabling better BBB penetrance, tumor uptake, and faster blood clearance than monoclonal antibodies [132,133]. Nanobodies are the single variable domain of the heavy-chain-only antibodies of *Camelidae* (camel, dromedary, llama, alpaca, vicuñas, and guananos) [63,134]. Nanobodies constitute the smallest molecules derived from antibodies (diameter of 2.5 nm and height of 4 nm; 15 kDa), although they still conserve full antigen-binding capacity with high specificity and affinity [135]. Nanobodies exert low toxicity and immunogenicity. Nanobodies have demonstrated their potential utility in diagnosing, monitoring, and therapy of a wide range of diseases [136,137]. Several differentially expressed proteins have been identified as glioblastoma targets with potential tumor-class predictive biomarker values [138,139]. Furthermore, a wide range of nanobodies targeting glioblastoma targets that have shown cytotoxic effects might constitute potential candidates for developing nanobody-based molecular imaging probes. Candidate nanobodies for immuno-PET approaches recognize molecular targets which play important roles in protein biosynthesis (TUFM, TRIM28), DNA repair and cell cycle (NAP1L1), and cellular growth and maintenance (EGFR, DPYSL2, β-Actin) [140,141,142]. Recently, a PD-L1-targeting nanobody-based tracer was evaluated to assess the changes in PD-L1 expression sensitively and specifically in different cancer types, which could help screen patients with high expression and guide PD-L1-targeting immunotherapies (Table 1) [112] (Figure 5b).

In contrast to conventional antibodies, nanobody-based immuno-PET probes may launch a novel era for the diagnosis of glioblastoma. Various molecular mechanisms for the transportation of nanobodies through the BBB have been extensively described [143,144,145,146] (Figure 6). Receptor-mediated transcytosis performs the movement of receptor ligands (e.g., transferrin, lactoferrin) across the BBB by a specific affinity-dependent unidirectional transport [147,148]. Nanobody FC5 (GenBank no. AF441486), the first nanobody described to traverse the BBB, binds the alpha(2,3)-sialoglycoprotein receptor in the brain endothelium [149,150]. FC5 set the basis for delivering BBB-impermeable therapeutic agents into the brain parenchyma by exploiting the receptor-mediated transcytosis of nanobodies [151]. Adsorptive-mediated transcytosis triggers the transport of basic molecules by electrostatic interactions with anionic microdomains on the cell membrane [152,153]. Several nanobodies with high isoelectric points (pI~9.5) have reported spontaneous delivery into the brain parenchyma. Basic nanobodies mVHH E9 (pI = 9.4), R3VQ (pI > 8.3), and A2 (pI > 9.5) have been shown to traverse the BBB and specifically label their molecular brain targets in vivo [154,155]. Transcytosis of nanobodies may be improved by other molecular shuttles such as peptide-decorated liposomes and cell-penetrating peptides (CPPs), which interact with the endothelial cells of the BBB and undergo nanobody internalization into the brain parenchyma [156,157,158].

In this regard, nanobodies crossing the BBB can be utilized as the targeting moieties of diagnostic and/or therapeutic immuno-PET tracers for CNS diseases. Nanobodies have already been used as non-invasive probes in several imaging techniques to visualize molecular pathologies, including glioblastoma [159]. First attempts labeled nanobodies with fluorescent dyes to perform in vivo optical imaging. The named EG2 nanobody and its bivalent (EG2-hFc) and pentavalent (V2C-EG2) formats were conjugated to the near-infrared (NIR) Cy5.5 fluorophore and successfully detected EGFRvIII expressing tumors in orthotopic mouse models of glioblastoma by NIR fluorescence imaging [160]. Similar results were obtained with the derivative nanobody EG2-Cys, labeled with NIR quantum dot Qd800 [161]. Cy5.5-labeled VHH 4.43, a nanobody directed against insulin-like growth factor-binding protein 7 (IGFBP7), was able to selectively detect blood vessels of glioblastoma after systemic injection in orthotopic glioblastoma bearing mice [162]. In addition, nanobodies have exhibited applicability as tracers in magnetic resonance imaging (MRI). Small unilamellar vesicles decorated with high Gd payload (Gd-DPTA), Cy5.5, and anti-IGFBP7 were used for dual (optical and MRI) in vivo imaging of glioblastoma orthotopic models [163]. Glioblastoma immuno-PET probes based on nanobodies targeting the hepatocyte growth factor (HGF) have demonstrated diagnostic potential in preclinical models. Nanobodies 1E2 and 6E10, linked to an albumin-binding nanobody (Alb8) and labeled with the positron emitter ^89^Zr, assessed HGF expression in xenografted glioblastoma mouse models [164]. These nanobody-based immuno-PET probes showed therapy potential as they delayed tumor growth. Other nanobody-based probes have evidenced diagnostic properties by performing MRI (R3VQ-S-(DOTA/Gd)_3_) [165] and micro-SPECT imaging ([^111^In]In-DTPA-pa2H [156]; ([^111^In]In-DTPA-pa2H-Fc [166]) of Alzheimer’s disease mouse models. These examples highlight the importance of the innovative field of immuno-PET tools based on the diagnostic potential of nanobodies for nuclear imaging and image-guided surgery [167].

Nanobodies have already evinced their clinical benefit in patients. In 2019, the Food and Drug Administration (FDA) and, more recently, the European Medicines Agency (EMA), approved the use of ALX-0681 (Caplacizumab; Ablynx NV, Ghent, Belgium) for adult patients with acquired thrombotic thrombocytopenic purpura [168,169]. ALX-0681 was the first nanobody reaching the clinic field, paving the way for a new era of diagnostics and therapeutics based on nanobodies. Nanobody-derived immuno-PET tracers are advancing through clinical trials. A human epidermal growth factor receptor 2 (HER2)-targeting nanobody ([^68^Ga]Ga-NOTA-anti-HER2 VHH1) has demonstrated its efficient diagnosis of primary breast carcinoma patients by PET/CT in a phase I study [170]. This nanobody-based tracer is being evaluated for the detection of breast-to-brain metastasis in a phase II trial (ClinicalTrials.gov NCT03331601). Recently, a phase I study was conducted to analyze the diagnostic potential of a ^99m^Tc labeled anti-PD-L1 nanobody ([^99m^Tc]Tc-NM-01) in non-small cell lung cancer patients by SPECT/CT imaging [171]. Nanobodies constitute a promising toolbox for innovative opportunities in the immuno-PET field towards personalized medicine.

## 7. Discussion

The current diagnosis of glioblastoma by conventional imaging methods presents multiple limitations. The most widely used technique in the primary diagnosis of this tumor is MRI [2,3]. While it provides very relevant anatomical information, it has a limited value for the diagnosis of glioblastoma because it mainly provides morphological information and does not allow proper discrimination of the tumoral tissue from concurrent processes such as inflammation, scar, edema, or bleeding that can lead to under or overestimate of the actual extension of the tumoral mass. Furthermore, some lesions, including brain access, lumps or space-occupying lesions, some pathologies coursing demyelination, the hemorrhagic transformation of stroke, or other lower-grade gliomas can be confounded with glioblastoma. The capacity of conventional MRI to differentiate tumor tissue from post-therapeutic effects following neurosurgical resection, radiation, alkylating chemotherapy, radiosurgery, and/or immunotherapy is also limited. Frequently, the discrimination between tumor recurrence and scar tissue is hard to determine by MRI after surgery [3]. As described above, other routinely neuroimaging techniques (CT, PET) used for the diagnosis of brain tumors also present multiple limitations.

While these imaging techniques are in continuous evolution and will benefit from the development of artificial intelligence [18,172] and bioinformatics, all these disadvantages highlight the importance of looking for other molecular-imaging-based technologies that allow the efficient and safe diagnosis of these brain tumors. Novel targeted imaging tools are needed to identify brain tumors at early stages, evaluate treatment response, and determine signs of progression or recurrence.

Despite continuous advances in the molecular classification [12,13,14] of gliomas, there have been no major improvements in patient survival over the past decades. There is an urgent need to integrate many clinically relevant alterations and biomarkers found in the multi-“omics” of cancer that are not yet included into the clinical management of glioblastoma patients due, in part, to the limited number of non-invasive biomarkers.

Current quantification of biomarkers in glioblastoma requires immunohistochemistry (IHC) and molecular biology analysis of surgical biopsies. However, this procedure is invasive and is not always feasible for all patients. Notably, the snapshot of a single biopsy usually does not capture the heterogeneity of glioblastomas, and several surgical biopsies and histopathological confirmation are required for a proper diagnosis. Some cases include cytology analysis of the cerebrospinal fluid (CSF); however, these techniques have limited sensitivity [173]. These hurdles represent a clinical challenge and an important risk for the patients leading to a lack of information about their glioblastomas.

Liquid biopsies are non-invasive tools that can provide longitudinal information about the tumor genomic landscape and facilitate the clinical management of patients. They analyze biomarkers present in the body fluids such as blood, urine, saliva, and CSF [174,175]. These biomarkers include circulating tumor cells, exosomes, and circulating tumor DNA (ctDNA) [174,175]. Detecting biomarkers in the blood is beginning a revolution that is transforming cancer diagnosis for multiple tumor types. However, the blood may not be a suitable source of ctDNA from patients with intracranial tumors, since ctDNA levels are infrequently detected in plasma. While ctDNA is detectable in the plasma of more than 75% of patients with advanced extracranial cancers, it is detectable in less than 10% of glioma patients [176,177].

The CSF is a source of ctDNA that can be sequenced and can reveal tumor heterogeneity providing diagnostic and prognostic information [173]. CSF-ctDNA liquid biopsies face multiple challenges including standardization of protocols, more extensive studies with more patients, and the implementation of well-designed and controlled clinical trials [173]. These hurdles need to be overcome to translate research findings into a tool for clinical practice [173].

Immuno-PET represents an attractive and innovative option for the diagnosis of gliomas allowing the analysis of biomarkers in a non-invasive manner. It combines the target selectivity and specificity of antibodies and subunit toward a biomarker with the high sensitivity, spatial resolution, and quantitative capabilities of PET. The development of novel immuno-PET tools will make it possible to conduct the non-invasive diagnosis and monitoring of patients over time using in vivo, quantifiable, 3D, whole body IHC [178], like a “virtual biopsy.” Immuno-PET will complement liquid biopsies to localize and characterize the gliomas and guide subsequent treatment decisions [179].

To date, several immuno-PET imaging tracers have been designed to target glioblastoma and have already proven successful in detecting gliomas in multiple preclinical models. These tracers target membrane proteins whose expression is altered in glioblastoma (including the EGFR, DLL4, EPHA2, CD47, AC133 antigen, and MT1-MMP) [94,95,96,97,98,99,100,101,106,107]: several components of the tumor microenvironment including vessels, macrophages, and extracellular matrix proteins [104,105,108,109,110,114]. Notably, [^89^Zr]Zr-DFO-fresolimumab, an immuno-PET tracer based on a monoclonal antibody that can neutralize all mammalian isoforms of TGF-β was assayed in humans, penetrated recurrent high-grade gliomas but did not result in clinical benefit [109]. Other immuno-PET tracers can serve to evaluate novel therapies [97,103,111,120] and to evaluate BBB disruption and permeability [108,114,125].

Targeted-radionuclide therapy is a strategy for the treatment of glioblastoma. This nuclear medicine approach enables the visualization of molecular biomarkers and pathways on a subcellular level using a biochemical vector coupled to a radionuclide that could work either for diagnosis (positron- or gamma-emitter) or for therapy (auger electrons-, β^—^- or α-emitter) [180]; when the radionuclides are used for the paired imaging and therapy agents, the strategy is called “radiotheranostics”. In the past years, targeted-radionuclide therapy has been used under a palliative context demonstrating to prolong overall survival, progression-free survival, and improve the patients´ life quality [181].

The administration of targeted-radionuclide therapy requires to confirm the presence of the glioblastoma target to determine on treatment options. To this end, immuno-PET emerges an opportunity as a possible quantitative imaging procedure to investigate the different biological properties and pharmacokinetics of tumor-targeted radiolabeled macromolecules including antibody fragments or engineered antibodies.

In addition to antibodies, multiple molecules (chemical, peptides, nanoparticles) have been designed to target specific biomarkers in the gliomas to develop probes of interest that can now be non-invasively imaged with multimodality molecular imaging techniques including MRI, CT, PET, single-photon emission computed tomography (SPECT), bioluminescence imaging, and near-infrared fluorescence to guide targeted therapies with a potential survival benefit and monitor patients’ response [182].

BBB permeability to the antibody-based probes remains a hurdle for immuno-PET applications. Glioblastomas are highly infiltrative and frequently alter the integrity of the BBB, resulting in leakiness, even though all glioblastomas may have clinically significant regions with an intact BBB [183]. These immuno-PET tracers could be informative to determine the grade of the BBB integrity of the tumor and could guide therapeutic interventions. The clinical realities of the contribution of the BBB to treatment failure in glioblastoma argue for renewed efforts to develop BBB-penetrating immuno-PET tracers.

The development of immuno-PET probes based on antibody subunits targeting glioblastoma biomarkers can overcome BBB selectivity emerging as promising probes for the non-invasive diagnosis of gliomas [131]. Among them, sdAbs such as nanobodies present multiple properties, including a smaller MW, enabling better BBB entrance, tumor uptake and biodistribution, and faster clearance than conventional antibodies [132,133].

Various molecular mechanisms for the transportation of nanobodies through the Blood–Brain Barrier (BBB) have been extensively described and include adsorptive and receptor-mediated transcytosis, somatic gene transfer, and the use of carriers or shuttles such as cell-penetrating peptides, extracellular vesicles, liposomes, and nanoparticles as well as device-based and physicochemical disruption of the BBB [143,144,145,146].

The development of nanobody-based radiotracers for the non-invasive diagnosis of glioblastoma by immuno-PET may also involve some potential challenges. First, suboptimal imaging of glioblastoma may be achieved due to the low penetration of nanobodies through the BBB [184], although some nanobodies have shown their potential to access the brain parenchyma [143]. Second, the administration of nanobodies with theragnostic potential may elicit immunogenic responses. The immunogenicity risk profile of nanobodies with potential clinical applications is being evaluated [185] and further humanization of the structure of nanobodies by genetic engineering techniques will minimize the activation of the immune system [72,73]. The administration of nanobodies may also lead to potential renal toxicity due to the high kidney uptake of nanobodies [186]. To solve this issue, several approaches have been developed to decrease renal uptake of nanobodies without inducing additional side effects: in vivo pretreatment with biomolecules (e.g., sodium maleate or fructose) [187] and PEGylation of nanobodies [188,189].

Furthermore, the production of nanobodies in the current good-manufacturing-practice (cGMP) grade is essential for their application in clinics. cGMP includes meeting preclinical quality standards, validating the nanobody format without tags utilized for production [190] and the site-specific radiolabeling of nanobodies, increasing the cost and time of their production [191,192]. Further improvement of image reconstruction and multimodal imaging approaches based on nanobodies will pave the way for a more precise diagnosis of glioblastoma by immuno-PET techniques.

Recently, a bivalent nanobody for the treatment of patients suffering from thrombotic thrombocytopenic purpura (Caplacizumab, ALX-0681) [168,169] received approval from the FDA and the EMA, representing a cornerstone for domain antibodies in the clinic and giving this area of research a boost. Nanobodies can be labeled with PET isotopes of shorter half-lives, such as ^68^Ga or ^44^Sc, which can be produced in a generator [79,80,81], making immuno-PET more accessible and affordable.

In contrast to a reduction in tumor size as observed in MRI, which usually represents late treatment effects, biomarker changes can occur earlier. Immuno-PET allows the quantification of biomarkers in a non-invasive manner in the whole body and holds the potential of detecting functional glioblastoma biomarker changes helping to an earlier diagnosis of glioblastoma, surveillance of patients, and monitoring of treatment response.

## 8. Conclusions

The current diagnosis of glioblastoma by MRI in some situations does not allow proper discrimination of the tumoral tissue from concurrent processes and can be confounding with other lesions and post-therapeutic effects [3].

Immuno-PET represents an attractive and innovative option for diagnosing gliomas, allowing the analysis of biomarkers in a non-invasive manner. By merging the target selectivity and specificity of antibodies and derivatives toward a biomarker with the high sensitivity, spatial resolution, and quantitative capabilities of PET [178], allowing the quantification of biomarkers in a non-invasive manner in the whole body.

To date, several immuno-PET imaging tracers have been designed to target glioblastoma and have already proven successful in detecting gliomas in multiple preclinical models, and they are advancing through clinical trials. The development of immuno-PET probes based on antibodies and nanobodies can overcome BBB selectivity emerging as promising probes for the non-invasive diagnosis, surveillance of patients, and monitoring of treatment response of gliomas [131].

## Figures and Tables

**Figure 1 cancers-14-00074-f001:**
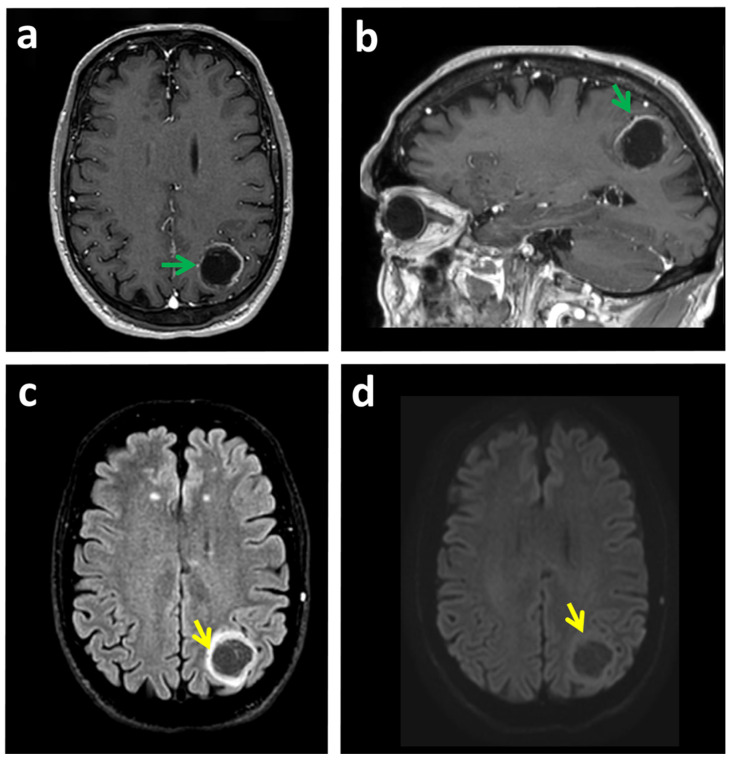
A case of glioma that could be confounded with brain access by MRI. (**a**,**b**) MRI images of a patient with glioblastoma in the left parieto-occipital lobe. T1W_3D-FFE MRI with gadolinium paramagnetic contrast. (**a**) Axial and (**b**) Sagittal reconstruction. The tumor shows contrast rim-enhancement (green arrow). This lesion was confounded with a brain abscess. (**c**) Fluid-attenuated inversion recovery (FLAIR) shows a parieto-occipital space-occupying lesion with peripheral hyperintensity and central hypointensity (yellow arrow). (**d**) The diffusion sequence shows minimal restriction of hydric diffusion (yellow arrow), which excludes the possibility that it is an abscess with typical behavior. Biopsy confirmed a diagnosis of glioblastoma.

**Figure 2 cancers-14-00074-f002:**
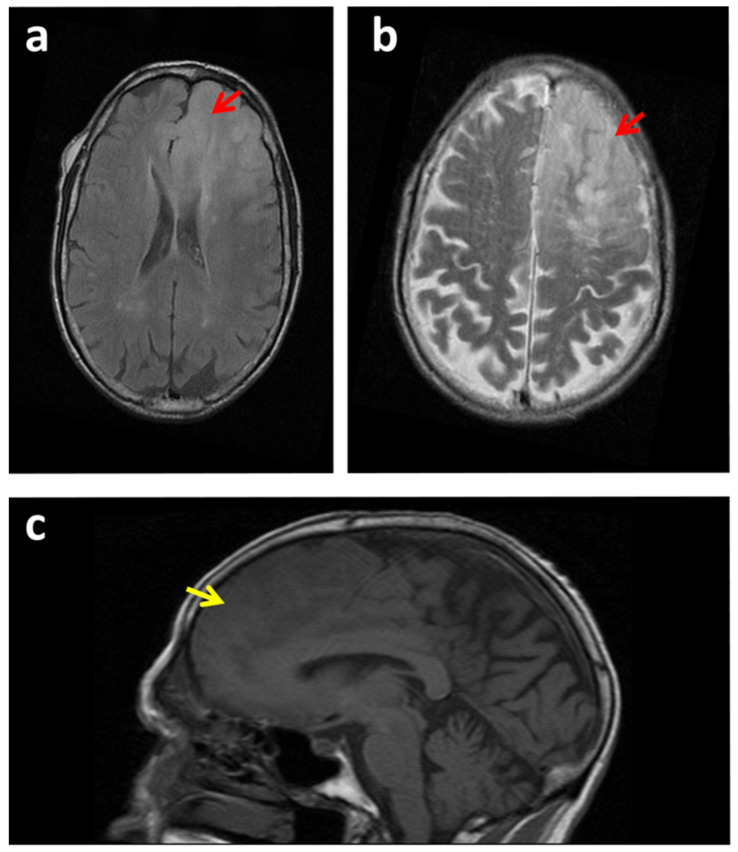
MRI scans of a case of glioma that could be confounded with an ischemic stroke. (**a**–**c**) MRI images of a patient with a glioma in the right frontal lobe (red arrows). (**a**) Inversion recovery fast spin-echo (IRFSE) Fluid-Attenuated Inversion Recovery (FLAIR), axial MRI. (**b**) Axial FSE T2 MRI image. (**c**) Spin-echo (SE) T1 sagittal MRI image. The space-occupying lesion could be confounded with an ischemic stroke in evolution (yellow arrows). Loss of gray and white matter differentiation. The lesion was confirmed to be a diffuse tumoral mass compatible with grade II astrocytoma by anatomopathological analysis.

**Figure 3 cancers-14-00074-f003:**
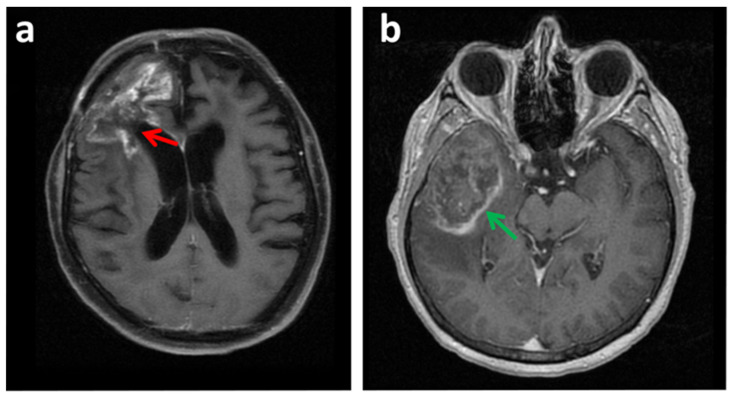
Other lesions can be confounding with glioblastoma. (**a**) Axial FS T1 MRI image with contrast of a glioblastoma recurrence (red arrow). In some situations, conventional MRI cannot correctly differentiate tumor tissue from post-therapeutic effects following neurosurgical resection and radiation. In this image, tumor recurrence was confounded with treatment necrosis produced by radiation. (**b**) Axial 3D Fast spoiled gradient echo (FSPGR) with MRI image. A patient suffering from hepatocellular carcinoma (HCC) presented one brain lesion detected by MRI (green arrow). In this situation a glioblastoma could be confounding with a brain metastasis. A biopsy indicated a glioblastoma and was discarded to be a brain metastasis from the HCC.

**Figure 4 cancers-14-00074-f004:**
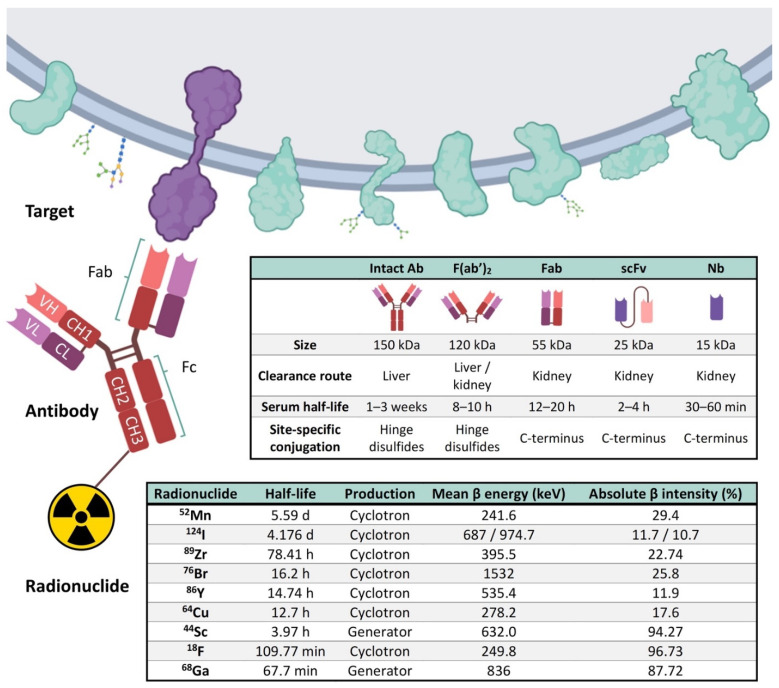
Representation of the three main components of the immuno-PET. Targets present in the external surface of the plasma membrane, antibody, and its derived immune fragments F(ab′)_2_, Fab, scFv, and Nb, and the most commonly used radionuclides are represented. A typical antibody (Immunoglobulin G, IgG) is composed of two heavy (H) chains and 2 light (L) chains. Heavy chains contain a series of immunoglobulin domains, usually with one variable domain (VH) that is important for antigen binding, and several constant domains (CH1, CH2, CH3). Light chains are composed of one variable (VL) and one constant (CL) domain. Abbreviations: Variable (V) and constant (C), Light (L), and Heavy (H); Ab, Antibody; Fab, Fragment antigen-binding; F(ab′)_2_,Fab dimer; scFv, single-chain variable fragment; Nb, Nanobody; ^18^F, Fluorine; ^44^Sc, Scandium; ^52^Mn, Manganese; ^64^Cu-Copper; ^68^Ga, Gallium; ^76^Br, Bromine; ^86^Y, Yttrium; ^89^Zr, Zirconium; ^124^I, Iodine [46,57,58]. Figure adapted with permission from Gónzalez-Gómez et al. [59]. Image created with BioRender.com (accessed on 6 September 2021).

**Figure 5 cancers-14-00074-f005:**
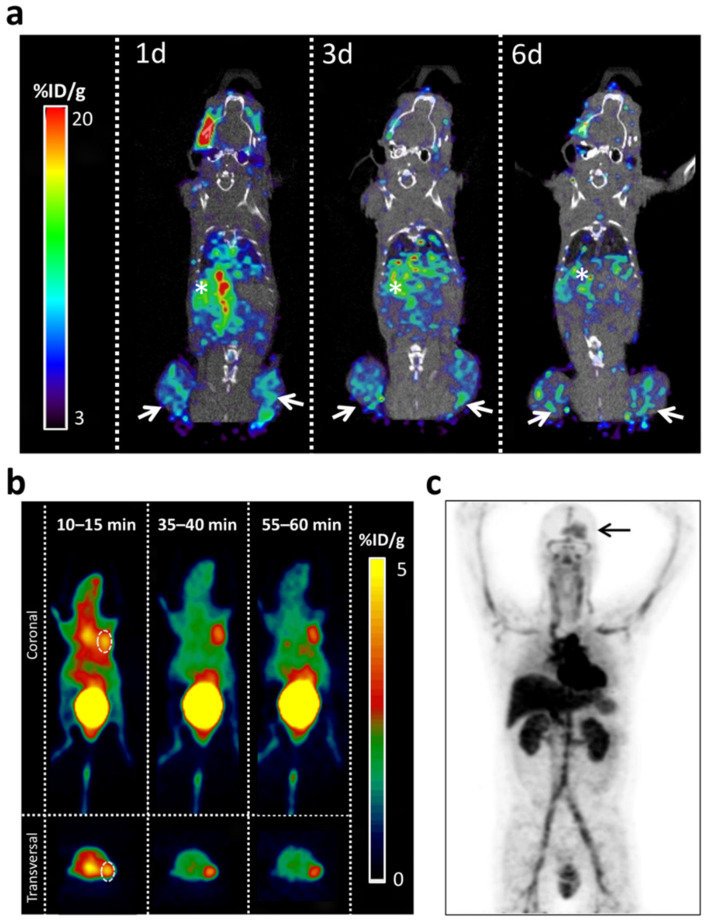
Examples of immuno-PET applications for the diagnosis of glioblastoma in preclinical models and patients. (**a**) PET/CT imaging with radiolabeled [^89^Zr]Zr-DFO-LEM 2/15 in a mouse bearing heterotopic xenografts containing patient-derived neurospheres. To generate subcutaneous heterotopic xenografts, 250,000 cells (MT1-MMP+, TS-543) were resuspended in 200 μL of a 1:1 mix of DMEM (Sigma, St. Louis, MO, USA) with Matrigel (BD Biosciences, San Jose, CA, USA). Next, the Matrigel:DMEM-cells mixture was injected subcutaneously into the flanks of 6 weeks athymic nude mice (Nude-Foxn1nu, Harlan Laboratories). Tumors were allowed to develop until palpable prior to immuno-PET analysis. Mice were inoculated with 2,3 MBq of [89Zr]Zr-DFO_LEM2/15 by retro-orbital sinus injection. (**a**–**c**) Representative fused PET/CT images. Sagittal whole-body sections at 1 (**1d**), 3 (**3d**) and 6 (**6d**) days post-injection. Images were obtained with a small-animal Argus PET-CT scanner (SEDECAL, Madrid, Spain). The PET studies (energy window 250–700 KeV and 30 min static acquisition) and CT (voltage 45 kV, current 150 μA, 8 shots, 360 projections and standard resolution) were performed at various time points post-injection in mice anesthetized by inhalation of 2–2.5% Isofluorane. The PET images were reconstructed using a 2D-OSEM (Ordered Subset Expectation Maximization) algorithm (16 subsets and two iterations), with random and scatter correction. Tissue activity is expressed as the percentage injected dose per gram of tissue (%ID/g). White arrows indicate the tumors’ location. White asterisk indicates the liver. Note the activity of the liver (asterisk) decreases gradually with time while it is maintained in the tumors (arrows). (**b**) MicroPET imaging of U87-MG xenograft model with [^68^Ga]Ga-NOTA-Nb109. Representative PET images obtained at different time points after injection. The tumor was denoted with a dotted line circle. Reprinted with permission from search was originally [112] 2021 Springer. The labeling of his panel was adapted for formatting. (**c**) Representative example of [^89^Zr]-Zr-DFO-fresolimumab PET on day 4 and uptake in brain tumor (arrow) in a human patient. Adapted with permission from ref. [109] 2015 SNMMI.

**Figure 6 cancers-14-00074-f006:**
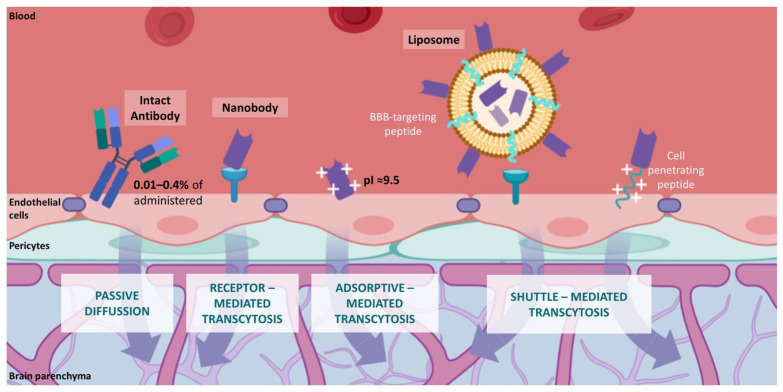
Molecular mechanisms of BBB permeability to antibodies. Comparison of conventional IgG antibodies (passive diffusion) and nanobodies (transcytosis mediated by BBB receptors, adsorptive processes, and BBB shuttle molecules). Image created with BioRender.com (accessed on 6 September 2021).

**Table 1 cancers-14-00074-t001:** Immune-PET tracers for glioblastoma.

PET Imaging Probes	Conjugation Strategy	Targets	Application	Models	References
[^18^F]AlF-NOTA/NODAGA-PODS-Z-EGFR:03115 (EGFR-targeting affibody molecule)	Cysteine-based random	EGFR	Many EGFR gene alterations have been identified in gliomas, especially glioblastomas.	Subcutaneous xenograft mouse model with U-87 MG vIII cells	[94]
[^124^I]I-PEG_4_-tptddYddtpt-ch806 (tptddYddtpt is a peptide ‘‘clicked″ onto dibenzyl- clooctyne(DBCO)-derivatized ch806)	Click chemistry	EGFR	ch806, an anti-EGFR mAb, can distinguish tumor cells with an amplified/overexpressed EGFR phenotype from normal cells having wild-type levels of EGFR expression.	Subcutaneous xenograft mouse model with U-87 MG.de2-7 cells	[95]
[^44^Sc]Sc−CHX-A″-DTPA−Cetuximab-Fab	Lysine-based random	EGFR	Radiolabeling and preclinical evaluation of ^44^Sc-labeled protein molecules.	Subcutaneous xenograft mouse model with U-87 MG	[96]
[^89^Zr]Zr-DFO-cetuximab	Lysine-based random	EGFR	^89^Zr-cetuximab was used to assess transient BBB disruption in vivo permeability induced by the combination of injected microbubbles with low intensity focused ultrasound.	Orthotopic murine glioma with GL261 cells	[97]
[^64^Cu]Cu-NOTA-Bs-F(ab)_2_ (bispecific immunoconjugate by linking two antibody Fab……fragments, an anti-EGFR and an anti-CD105)	Lysine-based random	EGFR and CD105	EGFR has been extensively studied as a target for anticancer therapy, and its activation stimulates tumor proliferation and angiogenesis. Similarly, CD105 (also called endoglin) is abundantly expressed on activated endothelial cells, and such over-expression is an adverse prognostic factor in many malignant tumor types.	Subcutaneous xenograft mouse model with U-87 MG	[98]
[^64^Cu]Cu-NOTA-EphA2-4B3 (human anti-EphA2 mAb)	Lysine-based random	EphA2	EphA2 receptor tyrosine kinase is overexpressed in several tumors, including glioblastoma.	Orthotopic brain glioblastoma murine models (two patient-derived cell lines and U-87 MG cells)	[99]
[^89^Zr]Zr-DFO-mCD47	Lysine-based random	CD47	CD47 is a membrane protein overexpressed on the surface of most cancer cells. It is involved in the increase in intracellular [Ca^2+^] that occurs upon cell adhesion to the extracellular matrix and is also a receptor for the C-terminal cell-binding domain of thrombospondin.	Orthotopic murine glioma with GL261 cells	[100]
[^64^Cu]Cu-NOTA-AC133 (anti-AC133 mAb)	Lysine-based random	AC133	AC133 is an N-glycosylation-dependent epitope of the second extracellular loop of CD133/prominin-1, a cholesterol-binding protein of unknown function that locates to plasma membrane protrusions. AC133^+^ tumor stem cells have been described for glioblastoma multiforme.	Orthotopic and subcutaneous xenograft mouse models with NCH421k and U-251 MG cells	[101]
[^89^Zr]Zr-DFO-bevacizumab (humanized anti-VEGF)	Lysine-based random	VEGF	^89^Zr-labeled bevacizumab was used to assess BBB opening with mannitol.	C3HeB/FeJ mice without tumors	[102]
[^68^Ga]Ga-DOTA-bevacizumab (humanized anti-VEGF)	Lysine-based random	VEGF	^68^Ga-labeled bevacizumab was used to assess BBB opening with focused ultrasound exposure in the presence of microbubbles.	Orthotopic murine glioma with U-87 MG cells	[103]
[^89^Zr]Zr-DFO-YY146 (anti-CD146 mAb)	Lysine-based random	CD146	CD146 plays an important role in several processes involved in tumor angiogenesis, progression, and metastasis. Its expression has been correlated with aggressiveness in high-grade gliomas.	Subcutaneous xenograft mouse model with U-87 MG and U251 cells	[104]
[^64^Cu]Cu-NOTA-YY146 (anti-CD146 mAb)	Lysine-based random	CD146	CD146 plays an important role in several processes involved in tumor angiogenesis, progression, and metastasis. Its expression has been correlated with aggressiveness in high-grade gliomas.	Orthotopic and subcutaneous xenograft mouse models with U-87 MG and U-251 MG cells	[105]
[^64^Cu]Cu-NOTA-61B (human anti-Dll4 mAb)	Lysine-based random	DII4	DII4 plays a key role to promote the tumor growth of numerous cancer types.	Subcutaneous xenograft mouse model with U-87 MG	[106]
[^89^Zr]Zr-DFO-LEM2/15 (anti-MM1-MMP mAb)	Lysine-based random	MT1-MMP/ MMP14	MMP14 is a metalloprotease frequently overexpressed in many tumors, and it is associated with tumor growth, invasion, metastasis, and poor prognosis.	Xenograft mice bearing human U251 cells and two orthotopic brain glioblastoma murine models (patient-derived TS-543 neurospheres and U-251 MG cells)	[107]
[^89^Zr]Zr-DFO-fresolimumab (human IgG4 mAb, 1D11)	Lysine-based random	TGFβ	TGFβ mediates extracellular matrix (ECM) remodeling, angiogenesis, and immunosuppression, and regulates tumor cell motility and invasion.	Orthotopic murine glioma with GL261 and SB28 cells	[108]
[^89^Zr]Zr-DFO-fresolimumab (human IgG4 mAb, 1D11)	Lysine-based random	TGFβ	TGFβ mediates ECM remodeling, angiogenesis, and immunosuppression, and regulates tumor cell motility and invasion.	Patients with recurrent high-grade glioma	[109]
[^89^Zr]Zr-DFO-F19 (anti-FAP monoclonal antibody)	Lysine-based random	FAP	FAP, a 170 kDa type II transmembrane serine protease, is expressed on glioma cells and within the glioma tumor microenvironment.	Subcutaneous xenograft mouse model with U-87 MG cells	[110]
[^89^Zr]Zr-DFO-PD-1	Lysine-based random	PD-1	^89^Zr labeled αPD-1 antibody was used to assess focal BBB permeability induced by high-intensity, focused ultrasound.	Orthotopic murine glioma with G48a cells	[111]
[^68^Ga]Ga-NOTA-Nb109 (anti-PD-L1 nanobody)	Lysine-based random	PD-L1	Evaluate the specific affinity of 68Ga-NOTA-Nb109 to several cancer cell lines that expressed endogenous PD-L1.	Subcutaneous xenograft mouse model with U-87 MG cells	[112]
[^89^Zr]Zr-DFO-169 cDb (anti-CD8 cys-diabody)	Lysine-based random	CD8	Proof-of-concept to detect CD8+ T cell immune response to oncolytic herpes simplex virus (oHSV) M002 immunotherapy in a syngeneic glioblastoma model.	Orthotopic syngeneic murine glioma with GSC005 cells	[113]
[^89^Zr]Zr-DFO-CD11b	Lysine-based random	CD11b	The most abundant population of immune cells in glioblastoma is the CD11b^+^ tumor-associated myeloid cells.	Mice bearing established orthotopic syngeneic GL261 gliomas	[114]
[^89^Zr/^177^Lu]Zr/Lu-Lumi804-CD11b	Lysine-based random	CD11b	Theragnostic approach for monitoring and reducing tumor-associated myeloid cells in gliomas to improve immunotherapy responses.	Mice bearing established orthotopic syngeneic GL261 gliomas	[115]
[^89^Zr]Zr-DFO-OX40	Lysine-based random	CD134	CD134 (or OX40) is an activated T-cell surface marker, known to be a costimulatory transmembrane molecule of TNF superfamily, primarily expressed on activated effector T cells and regulatory T cells.	Mice bearing established orthotopic GL261 gliomas	[116]

Abbreviations: CD8—Cluster of differentiation 8; CD11b—Integrin αM; CD47—Cluster of differentiation 47; CD105—endoglin; CD134—Tumor necrosis factor receptor superfamily, member 4 (TNFRSF4); CD146—Cluster of Differentiation 146; DLL4—Delta-Like Ligand 4; EGFR—Epidermal Growth Factor Receptor; EPHA2—Ephrin type-A receptor 2; FAP—Fibroblast activation protein alpha; MT1-MMP/MMP14—Membrane-type 1 matrix metalloproteinase; PD-1—programmed cell death receptor-1; PD-L1—Programmed cell death ligand 1; TGFβ—Transforming growth factor β; VEGF—Vascular Endothelial Growth Factor.

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
