# Peer review of "Diagnosis of Glioblastoma by Immuno-Positron Emission Tomography"

_cancers, 2021, doi:10.3390/cancers14010074_

Round 1

Reviewer 1 Report

Overall this is an interesting summary of immunoPET for glioma diagnosis. The manuscript follows a sound logic to demonstrate the importance and current diagnostic landscape of glioblastoma, then introduce and discuss immunoPET and new tracers for this purpose. The reviewer supports the publication of this manuscript. A few comments are listed below:

  1. The “Introduction” ends with current methods to treat glioma, but the next part “Current status of glioblastoma classification and diagnosis” focuses on the classification and did not touch treatment any more. The transition can be smoother.
  2. The authors did a great job on preparing schemes. More immunoPET images can be added to better demonstrate the merits.
  3. The nomenclature of immunoPET tracers should be unified: three different expressions, [89Zr]Zr-DFO-LEM 2/15; pa2H-DTPA-111In, and 68Ga-NOTA-anti-HER2 VHH1, have all been used in the main text.
  4. Potential challenges of immunoPET tracer development can be listed in the discussion, for example their GMP production, quality control, potential toxicity or immunogenicity, and radiolabeling stability and reproducibility.
  5. Typos should be modified throughout the manuscript, for example, on line 553, “99mTc”.

Reviewer 2 Report

The authors present a review on advanced metabolic techniques PET-based for the diagnostic work-up of glioblastomas. 

This is a well-written article; its point of strength is the focus on “immunotargeted imaging”. I think that it could represent a valid update for the clinical management of GBM, but I have several issues that must be clarified. This aspect is valid also for the first Introduction paragraph of the Introduction (lines 1-45, ref 1-5)

Line 38: for the epidemiological description, it is better to use a more updated reference. CBTRUS is the main resource for the epidemiology of CNS tumors (see the last version, Ostrom QT et al. CBTRUS Statistical Report: Primary Brain and Other Central Nervous System Tumors Diagnosed in the United States in 2014-2018. Neuro Oncol. 2021 Oct 5;23(12 Suppl 2):iii1-iii105. doi: 10.1093/neuonc/noab200)

Fig 1: whereas it is true that brain abscesses should be confounded with GBM, DWI/ACD or perfusion greatly help in distinguishing between them. Authors should improve the figure with these sequences, otherwise, it looks that the diagnostic value of MRI is limited

Lines 165-166 "Furthermore, recent studies indicate that 60% of rebound glioblastomas occur at the biopsy site [24,25]"; whereas these 2 studies are not really recent, I don' fully agree with the sentence. The recurrency of a GBM after macroscopically removed surgery usually happens at the boundaries of the surgical cavity. If the authors, instead, mean only stereotactic or frameless biopsies, it is not a "rebound" disease but only its natural history.

Linse 185-188: what about the possible application of Choline? (see i.e. Alongi P et al. Choline-PET/CT in the Differential Diagnosis Between Cystic Glioblastoma and Intraparenchymal Hemorrhage. Curr Radiopharm. 2019;12(1):88-92. doi: 10.2174/1874471011666180817122427. PMID: 30117406 AND Laudicella R et al. Unconventional non-amino acidic PET radiotracers for molecular imaging in gliomas. Eur J Nucl Med Mol Imaging. 2021 Nov;48(12):3925-3939. doi: 10.1007/s00259-021-05352-w. Epub 2021 Apr 13)

Round 2

Reviewer 2 Report

The authors almost fully addressed all my issues; I suggest including the new version of figure 1 rather than the previous one in the definitive version.

AFter this correction, I think that the manuscript is suitable for publication